

# Timescale-dependence of the relationship between the East Asian summer monsoon strength and precipitation over eastern China in the last millennium

Jian Shi[1], Qing Yan[2,3], Huijun Wang[2,3]

[1]College of Atmospheric Science, Nanjing University of Information Science and Technology, Nanjing, 210044, China.
[2]Nansen-Zhu International Research Centre, Institute of Atmospheric Physics, Chinese Academy of Sciences, Beijing, 100029, China.
[3]Collaborative Innovation Center on Forecast and Evaluation of Meteorological Disasters, Nanjing University of Information Science and Technology, Nanjing, 210044, China.

*Correspondence to*: Qing Yan (yanqing@mail.iap.ac.cn)

**Abstract.** Precipitation/humidity proxies are widely used to reconstruct the historical East Asian summer monsoon (EASM) variation based on the assumption that summer precipitation over eastern China is closely and stably linked to the strength of EASM. However, whether the observed EASM-precipitation relationship (e.g., increased precipitation with a stronger EASM) was stable throughout the past time remains unclear. In this study, we used model outputs from the Paleoclimate Modelling Intercomparison Project Phase III and Community Earth System Model to investigate the stability of the EASM-precipitation relationship over the last millennium on different timescales. The model results indicate that the EASM strength (defined as the regionally averaged meridional wind) enhanced in the Medieval Climate Anomaly (MCA; ~950–1250 A.D.), during which there was increased precipitation over eastern China, and weakened during the Little Ice Age (LIA; ~1500–1800 A.D.), during which there was decreased precipitation, consistent with precipitation/humidity proxies. However, the simulated EASM-precipitation relationship is only stable on a centennial and longer timescale and is unstable on a multi-decadal timescale. The nonstationary multi-decadal EASM-precipitation relationship broadly exhibits a quasi-60-year period, which may be attributed to the internal variability of the climate system and have no significant correlation to external forcings. Our results have implications for understanding the discrepancy among various EASM proxies on a multi-decadal timescale and highlight the need to rethink reconstructed decadal EASM variations based on precipitation/humidity proxies.

## 1 Introduction

The East Asian summer monsoon (EASM) is a crucial component of Asian monsoons, consisting of tropical and subtropical components (Tao and Chen, 1987; Wang et al., 2008). The main characteristic of EASM is the prevailing southerlies in the lower-troposphere over the East Asia region (Fig. 1a) induced by the pressure contrast between the East Asia continent and western North Pacific (Li et al., 1996; Shi et al., 1996). As the EASM strengthens (weakens), summer precipitation over eastern China generally increases (decreases) due to sufficient (deficient) water vapor being transported from tropical and



subtropical oceans (e.g., Lau et al., 1988; Ding et al., 2008). The assumption that the observed relationship between the EASM strength and summer precipitation over eastern China (i.e., EASM-precipitation relationship) was stable throughout the past time is the presupposition for paleo-EASM reconstruction using precipitation/humidity-relevant proxies, considering the difficulties in directly reconstructing winds and atmospheric circulations. However, previous studies indicate that the

5 aforementioned EASM-precipitation relationship is possibly changeable over recent decades (e.g., Shi and Zhu, 1998; Li et al., 2010; Zhang, 2015) and in the future warming world (e.g., Xin et al., 2013; Ren et al., 2016). Thus, it is necessary to investigate the stability of the EASM-precipitation relationship during the past time, which may shed light on the reliability of paleo-EASM reconstructions.

The last millennium, in which large amounts of historical documents and proxies are available to reconstruct the EASM,

provides an opportunity to explore variations in the EASM on different timescales. According to the temperature reconstructions (e.g., Crowley et al., 2000; Mann et al., 2008; Cook et al., 2013), there were two typical climate anomalous periods over the last millennium: the Medieval Climate Anomaly (MCA; approximately 950–1250 A.D.) and Little Ice Age (LIA; approximately 1450–1850 A.D.). The MCA/LIA allows us to study the EASM response in a warmer/colder climate background. On a centennial timescale, Wang et al. (1987) found that floods occurred frequently in North (South) China in

the MCA (LIA) based on historical documents, indicating a stronger (weaker) EASM at that time. Using oxygen isotope records from stalagmites in central China, Zhang et al. (2008) also showed that the EASM was enhanced (decreased) in the MCA (LIA), as confirmed by other stalagmite records in central and North China (e.g., Tan et al., 2009; 2011), as well as several pollen and lake sediments records (e.g., Meng et al., 2009; Liu et al., 2011). Additionally, by combing multiple proxy records, Chen et al. (2015) indicated wetter (drier) conditions over North China during the MCA (LIA), implying a stronger

(weaker) EASM (Shi et al., 2016b). Nevertheless, on a shorter timescale, there are obvious discrepancies among various EASM records (e.g., Paulsen et al., 2003; Zheng et al., 2006; Hu et al., 2008; Zhang et al., 2008; Tan et al., 2009; Wan et al., 2011). For example, the stalagmite record described in Zhang et al. (2008) showed a weaker EASM and drier period over the 860–940 A.D. period, while Zheng et al. (2006) suggested that eastern China was generally wetter based on historical documents. Moreover, Zhang et al. (2010) implied that several EASM stalagmite records do not match well with the

observed EASM variation over the past 150 years on an interdecadal timescale.

These proxies generally indicate a stronger (weaker) EASM in the MCA (LIA) with enhanced (reduced) precipitation over eastern China on a multi-centennial timescale, but exhibit large discrepancies on a shorter timescale. Given the sporadically dispersed proxies and limited environmental variables derived from them, the inconsistency among various EASM records is not yet fully understood. Differences in the local climate between each proxy site and uncertainty in the interpretation of

30 proxies may contribute to the aforementioned data discrepancy (Tan et al., 2007; Chen et al., 2015). However, the evolution of the EASM in the last millennium has been broadly reconstructed using precipitation/humidity-relevant proxies (e.g., Zheng et al., 2006; Zhang et al., 2008). Therefore, the research question is whether the inconsistency (consistency) can be partially attributed to the decoupling (coupling) of East Asian summer winds and precipitation over eastern China on different timescales.



Numerical simulations offer a useful tool to examine climate change and the associated dynamic mechanisms in the past. Based on the ECHO-G simulation, Liu et al. (2011) indicated that the EASM was strengthened and summer precipitation increased over eastern China during the MCA, which were attributed to the increased total solar irradiation. Man et al. (2012) indicated that the EASM became stronger during the MCA, but precipitation increased (decreased) in North (South) China in

the MPI-ESM simulations. Peng et al. (2014) also implied that several severe droughts that occurred over eastern China were associated with a weakened EASM. Shi et al. (2016a) analyzed the Asian circulation and precipitation changes between the MCA and LIA using Paleoclimate Modelling Intercomparison Project Phase III (PMIP3) last millennium simulations. Nevertheless, existing modeling studies have paid little attention to the stability of the EASM-precipitation relationship during the last millennium—which is of vital importance for reconstructing EASM evolution with precipitation-relevant

proxies and is the focus of this study.

Here, we use the last millennium experiments of the PMIP3 models and Community Earth System Model (CESM) to study changes in the EASM and precipitation over eastern China over the last millennium (~850–1850 A.D.). We focus on the spatial-temporal stability of the EASM-precipitation relationship on different timescales. Understanding these issues will improve our cognition of the rationality of applying precipitation/humidity-relevant proxies as indicators for EASM changes.

The remainder of this study is organized as follows. In Sect. 2, we introduce the model data and methods used in our analysis. In Sect. 3, we discuss the variation of the EASM strength and summer precipitation over eastern China over the last millennium. In Sect. 4, we investigate the stability of the EASM-precipitation relationship and possible drivers of this changeable relationship. In Sect. 5, we summarize the main results and uncertainties.

## 2 Data and Methods

We use the model outputs of the PMIP3 last millennium simulations from nine climate models (Table 1), excluding MIROC-ESM because of its climate drift in long-term simulations (Gupta et al., 2013). These simulations have a rough time span of 850–1850 A.D. and are mainly forced by total solar irradiance, volcanic eruptions, land use/cover, and greenhouse gases. In addition, we use the CESM last millennium ensemble (CESM-LME) simulations (Otto-Bliesner et al., 2016) to examine the roles of different forcings in the EASM-precipitation relationship. The CESM-LME uses CESM 1.1 with the Community

Atmosphere Model version 5 (CAM5) as its atmosphere component (Hurrell et al., 2013), which features a ~2° horizontal resolution. The external forcings applied in the CESM-LME simulations follow those of the CCSM4 in the PMIP3 last millennium experiment. In particular, we analyze nine CESM-LME full-forcing experiments and numbers of sensitivity experiments, including one control experiment, four solar activity experiments, five volcanic eruption experiments, and three greenhouse gases, land use and Earth's orbit experiments.

Modern observations show that the summer (June–July–August) prevailing winds over East Asia (i.e., the EASM) feature southerlies blowing from tropical oceans and the North Pacific (Fig. 1a). Thus, we define the EASM strength as the regionally averaged 850-hPa meridional winds over East Asia (20°–45°N, 105°–135°E) in summer (Jiang et al., 2013). The



running pattern correlation (RPC) is applied to reveal the spatial stability of the EASM-precipitation relationship throughout the last millennium. Specifically, we calculate the geological distributions of the correlation between the EASM strength and summer precipitation over East Asia (10°–50°N, 100°–130°E) over an α-year window period (α is the running window size) and the full period (~850–1850 A.D.), and define their pattern correlation as the RPC. We select the MCA as 950–1250 A.D.

and LIA as 1500–1800 A.D. with an identical time span.

## 3 Variations of EASM and precipitation over eastern China during the last millennium

We first evaluated the performance of PMIP3 models in reproducing modern summer 850-hPa meridional winds over East Asia using the Taylor diagram (Taylor, 2001). As shown in Fig. 1b, most PMIP3 models can simulate the observed summer southerly winds over East Asia, except for CSIRO-Mk3L-1-2, which has a negative spatial correlation to observations and

10 hence is excluded. The pattern correlations between the remaining eight models and observations vary from 0.32 (MRI-CGCM3) to 0.79 (GISS-E2-R), all passing the 95% significance test. The centered root-mean-square errors range from 0.99 (MRI-CGCM3) to 1.55 (HadCM3), indicating that the PMIP3 models generally produce a slightly stronger EASM than observations. Thus, we use the eight models shown in Fig. 1b in our further analyses.

The multi-model ensemble mean (MEM) shows that EASM are generally stronger during the MCA and weaker during the

15 LIA (Fig. 2a). The majority of the PMIP3 models (six of eight models) support the MEM result, while BCC-CSM1-1 and FGOALS-s2 simulate a weaker (stronger) EASM during the MCA (LIA) (Fig. 2c). Correspondingly, summer precipitation over eastern China (20°–45°N, 105°–120°E) increases in the MCA and decreases in the LIA based on the MEM. The spatial pattern of summer winds and precipitation anomalies between the MCA and LIA (Fig. 2d) further indicate that when southerlies over East Asia reinforce, precipitation over almost all of eastern China increases, consistent with the findings of

20 Liu et al. (2011). In addition, the in-phase changes of the EASM strength and precipitation over eastern China are shown in both the MEM and individual PMIP3 models results (Fig. 2c), implying a robust positive EASM-precipitation relationship on a multi-centennial timescale.

Moreover, the MEM result, which shows an enhanced EASM and associated increased summer precipitation during the MCA relative to the LIA, is broadly consistent with the reconstructed EASM derived from precipitation/humidity records in

the East Asian monsoon region (e.g., Zhang et al., 2008; Tan et al., 2011). The possible mechanisms for EASM and precipitation changes are as follows. The PMIP3 models simulate a significant warming (cooling) in the Asia continent and weaker warming (cooling) in adjacent oceans during the MCA (LIA) in summer (Fig. 3a); this results in an enhanced (decreased) land–sea pressure contrast and thus a stronger (weaker) EASM. Meanwhile, the meridional temperature gradient reduces (increased) in the middle to high latitude and leads to an overall weakening (strengthening) in the upper westerlies

(Fig. 3b), which is favorable for a stronger (weaker) EASM (Zhou and Yu, 2005). These results are similar to previous findings of single model studies (e.g., Liu et al., 2011; Man et al., 2012; Shi et al., 2016b).



To summarize, although amplitude and spatial changes in the EASM and precipitation vary largely among the PMIP3 models in the last millennium, the majority of these models indicate a strengthened EASM and enhanced summer precipitation over eastern China during the MCA relative to the LIA. Hence, from the perspective of PMIP3 simulations, using the precipitation/humidity-relevant proxies is convincing to indicate the EASM changes between the MCA and LIA.

## 4 The EASM-precipitation relationship during the last millennium

In this section, we discuss whether the EASM-precipitation relationship (i.e., enhanced summer precipitation over eastern China with a stronger EASM) maintained throughout the last millennium, that is, we discuss the stability of the EASM-precipitation relationship. After evaluating the EASM strength in the PMIP3 models (Fig. 1), we further assessed their performance in reproducing the modern EASM-precipitation relationship (Fig. 4). The observations show that a strengthened EASM brings sufficient moisture from adjacent oceans and forms abundant summer precipitation over eastern China, especially in the middle to high latitude regions (Fig. 4a). The geological distribution of the correlation between the EASM strength and summer precipitation simulated by the PMIP3 models (with the exception of the FGOALS-s2) is roughly similar to the observed distribution, with spatial correlation coefficients ranging from 0.17 to 0.47 (all passing the 95% significance test). FGOALS-s2 simulates no significant precipitation changes over eastern China in response to EASM strength changes and hence is eliminated from the following discussions. Eventually, we selected the remaining seven PMIP3 models to study the EASM-precipitation relationship over the last millennium.

### 4.1 Stability of the EASM–precipitation relationship over the last millennium

We apply a 31-year and 101-year running correlation to represent the EASM-precipitation relationship on a short timescale (i.e., multi-decadal) and long timescale (i.e., centennial timescale), respectively. On each timescale, we consider two aspects: (1) the stability of the spatial pattern of the correlation between the EASM strength and eastern China summer precipitation and (2) the stability of the relationship between the EASM strength and regionally averaged summer precipitation over the middle latitudes of eastern China (25°–45°N,105°–120°E), where precipitation changes are more positively related to the EASM variation in the observation and PMIP3 models (Fig. 4).

First, we calculate the RPCs on different running window sizes to represent the spatial stability of the EASM-precipitation relationship on each timescale. As shown in Fig. 5, on a multi-decadal timescale, time-averaged RPCs vary from 0.54 (BCC-CSM1-1 and CCSM4) and 0.77 (HadCM3) and pass the 95% significance test in the PMIP3 simulations. However, the RPCs show large fluctuations in almost all of the PMIP3 models and even fall to near-zero, with standard deviations ranging from 0.09 (HadCM3) to 0.18 (MPI-ESM-P) over the last millennium. The RPCs in the GISS-E2-R and HadCM3 simulations are relatively more stable than in other models. This result indicates that the EASM-precipitation relationship may change over time on a multi-decadal timescale (e.g., Fig. S1), which potentially reduces the reliability of precipitation-relevant proxies located in eastern China in representing the EASM strength on a short timescale. Moreover, there are no coherent time



intervals for high and low RPCs among the PMIP3 models, suggesting that the variability of the EASM-precipitation relationship may be not induced by external forcings. Nevertheless, on a centennial timescale, RPCs are more stable than those on a multi-decadal timescale in all cases, with averages larger than 0.79 and standard deviations smaller than 0.07 among individual PMIP3 models. Such a stable EASM-precipitation relationship gives us confidence in reconstructing the centennial EASM records with precipitation-relevant proxies.

The aforementioned results are further confirmed by running correlations (RCs) between the EASM strength and regionally averaged summer precipitation over the middle latitudes of eastern China (Fig. 6). On a multi-decadal timescale, the RCs are highly time-dependent. Although the time-averaged RCs pass the 95% significance test, they are insignificant and even negative for many time intervals over the last millennium in the majority of the PMIP3 models. The results of the GISS-E2-R and HadCM3 simulations show relatively more stable RCs, similar to those revealed by the RPCs. On a centennial timescale, the RCs pass the significance test for almost the entire period, indicating a more stable, closer EASM-precipitation relationship.

## 4.2 Reasons for fluctuations in the multi-decadal EASM-precipitation relationship

We found that the unstable multi-decadal EASM-precipitation relationship has obvious periodicity. The power spectrum results (Fig. 7) indicate that both the 31-year RPCs and RCs show a period of approximately 60 years (~50–80 years) in the majority of the PMIP3 last millennium simulations. This periodicity might be attributed, to a certain extent, to the multi-decadal variability in summer precipitation over eastern China, which exhibits an intense multi-decadal periodicity in the PMIP3 models (Shi et al., 2016a). Multi-decadal variability is also notable in precipitation/humidity records located over eastern China (e.g., Zhu et al., 2002; Zheng et al., 2006; Li et al., 2011), which supports our inferences from the PMIP3 simulations.

We further use the CESM-LME experiments to determine potential drivers for fluctuations in the multi-decadal EASM-precipitation relationship. Similar to most PMIP3 models, CESM broadly reproduces the observed EASM and EASM-precipitation relationship, especially over North China (Fig. S2). The CESM-LME full-forcing experiments indicate that the EASM-precipitation relationship (expressed by the RCs between the EASM strength and summer precipitation over the middle latitudes of eastern China) is stable on a centennial timescale and unstable on a multi-decadal timescale (Fig. 8), which has a period of almost 60 years in most ensemble members (Fig. 9).

The CESM sensitivity experiments further demonstrate that this periodicity is possibly induced by the internal variability of the climate system. An almost 60-year period in the multi-decadal EASM-precipitation relationship occurs in the CESM-LME control experiment (Fig. 10a), in which no external forcing is applied. An approximately 120-year period also exists, but is not common in its full-forcing experiment members (Fig. 9). Meanwhile, the near 60-year period appears in most CESM-LME single-forcing experiments (Fig. 10b–10s), further emphasizing the crucial role of internal variability in forming this periodicity. More importantly, among the same single-forcing experiments, there are no common periods corresponding to the applied external forcing. For example, the EASM-precipitation relationships do not consistently exhibit



the solar periods, such as quasi-88 or quasi-120 years, in four solar-forcing experiments (Fig. 10b–10e). This result suggests that the external forcings may not be important for the unstable EASM-precipitation relationship on a multi-decadal timescale.

We propose a possible driver for the changeable EASM-precipitation relationship according to the CESM-LME simulations.
Figure 11 shows that in the majority of the CESM-LME full-forcing experiments, the EASM-precipitation relationship tends to be closer when the sea surface temperature (SST) over North Atlantic is colder on a multi-decadal timescale, resembling a negative Atlantic Multi-decadal Oscillation (AMO) pattern. This implies that the alternation of the weak–strong EASM-precipitation relationship may be partially related to the phase changes of AMO. Although it is not apparent in the CESM-LME simulations, another noteworthy basin-scale SST forcing is the Pacific Decadal Oscillation (PDO), which has a similar multi-decadal periodicity to that of the AMO. Previous studies have indicated that the AMO and PDO, as well as combinations of their different phases, may play important roles in modulating summer circulation and precipitation over eastern China on a multi-decadal timescale (e.g., Lu et al., 2006; Yu et al., 2015; Lin et al., 2016; Shi et al., 2016b; Yang et al., 2017). Nevertheless, it remains unclear how these oceanic drivers influence the EASM-precipitation relationship. This issue is difficult to interpret due to the complexity of the formation of summer precipitation over eastern China, which is not only dependent on water vapour transported by the EASM, but is also attributed to local ascending motion (e.g., Ding et al., 2008; Zhu et al., 2011) and cold air activities (Huang et al., 2014; Lu et al., 2014) among other factors. Given the short time span of the observations and weak ability of existing climate models to simulate the internal variability of the climate system, the physical processes remain elusive.

Overall, simulations of the last millennium generally reveal that the stability of the EASM-precipitation relationship depends on the timescale. On a shorter timescale, (at most the multi-decadal timescale), precipitation over eastern China is not always significantly influenced by the EASM strength, which may be due to the internal variability of the climate system. Nevertheless, on a longer timescale, the EASM-precipitation relationship is much more stable.

## 5 Discussion and conclusions

In this study, the EASM-precipitation relationship is examined in simulations of PMIP3 and CESM-LME over the last millennium (~850–1850 A.D.). CSIRO-Mk3L-1-2 fails to capture the climatological EASM, and FGOALS-s2 shows a weak performance in reproducing the observed EASM-precipitation relationship. After ruling out these two models, the ensemble of the remaining "better-performance" models shows that EASM strengthened during the MCA and weakened during the LIA, consistent with geological evidence. The enhanced land–sea thermal contrast and weakened upper westerlies are the main causes for the EASM strength variation during the MCA relative to the LIA. In all of the PMIP3 models, changes in summer precipitation over eastern China are positively related to EASM strength changes.

However, we suggest that the EASM-precipitation relationship is timescale-dependent according to the climate models. On a centennial timescale, eastern China summer precipitation is significantly and stably influenced by the EASM strength, in

agreement with simulated EASM and precipitation changes between the MCA and LIA. By contrast, on a multi-decadal timescale, the EASM-precipitation relationship is non-stationary over the last millennium and fluctuates with a period of approximately 60 years. Further investigations using CESM-LME control and single-forcing experiments suggest that this periodicity of the multi-decadal EASM-precipitation relationship is an internally forced result but is not externally forced.

5 The uncertainties of our results likely arise from the following aspects. Foremost, the modern EASM-precipitation relationship is actually very complex, as there are many EASM indices and different patterns of the summer rainbelt over eastern China. In this study, we mainly focus on the overall strength of EASM and its relationship to the entire amount of precipitation over eastern China. Another inevitable reason is the limited ability of models to reproduce the modern East Asian climate. We make use of as many climate models or ensembles as possible to minimize the potential deviation of 10 individual models.

Though uncertainties exist, this study investigates the timescale-dependent stability of the EASM-precipitation relationship, which may be meaningful for interpreting climatic information derived from humidity/precipitation proxies. It could be inferred from our analysis that the multi-decadal variability of EASM, which cannot be ignored in the last millennium, is hard to reconstruct with humidity/precipitation proxies, at least during periods with a weak EASM-precipitation relationship. 15 In other words, no matter how high temporal resolution the proxies have, it is likely that they can only derive EASM variations on a centennial or longer timescale. The multi-decadal or shorter timescale components of proxy records may reflect local humidity changes, which provides a possible explanation for the discrepancies among proxies representing EASM variation on short timescales.

20 **Data availability.**

The PMIP3 last millennium simulation data can be obtained in the Program for Climate Model Diagnosis and Inter-comparison (PCMDI; http://pcmdi9.llnl.gov) archives and the CESM-LME data are available at http://www.cesm.ucar.edu/projects/community-projects/LME/data-sets.html.

**Competing interests.**
25 The authors declare that they have no conflict of interest.

**Acknowledgements.**
We acknowledge the climate modeling groups of the PMIP project and the CESM-LME project for sharing their model output. This research was supported by the National Science Foundation of China (Grant No. 41421004 and 41772179), the External Cooperation Program of BIC, Chinese Academy of Sciences (Grant No. 134111KYSB20150016), and Chinese 30 Academy of Sciences-Peking University (CAS-PKU) partnership program.



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



**Table 1. PMIP3 model simulation information and their main forcing.**

| Model | Country | Atmosphere resolution | Main forcing | |
|---|---|---|---|---|
| | | | Solar | Volcanic |
| BCC-CSM-1.1 | China | $128 \times 64$, L26 | Vieira et al. (2011); Wang et al. (2005) | Gao et al. (2008) |
| CCSM4 | USA | $288 \times 192$, L26 | Vieira et al. (2011) | Gao et al. (2008) |
| CSIRO-Mk3L-1-2 | Australia | $64 \times 56$, L18 | Steinhilber et al. (2009) | Crowley et al. (2008) |
| FGOALS-s2 | China | $128 \times 60$, L26 | Vieira et al. (2011); Wang et al. (2005) | Gao et al. (2008) |
| GISS-E2-R | USA | $144 \times 90$, L40 | Vieira et al. (2011) Wang et al. (2005) | Gao et al. (2008) |
| HadCM3 | UK | $96 \times 73$, L19 | Steinhilber et al. (2009); Wang et al. (2005) | Crowley et al. (2008) |
| IPSL-CM5A-LR | France | $96 \times 95$, L39 | Vieira et al. (2011); Wang et al. (2005) | Gao et al. (2008) |
| MPI-ESM-P | Germany | $196 \times 98$, L47 | Vieira et al. (2011); Wang et al. (2005) | Crowley et al. (2008) |
| MRI-CGCM3 | Japan | $320 \times 160$, L48 | Delaygue and Bard (2009); Wang et al. (2005) | Gao et al. (2008) |





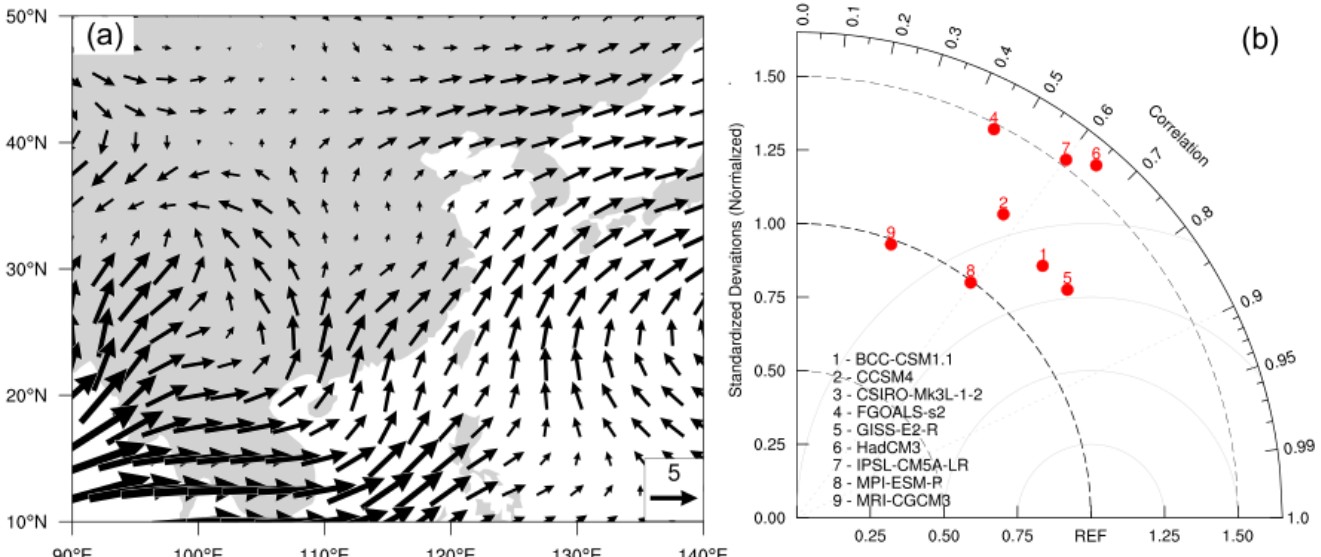

**Figure 1. (a) Climatological (1979–2000 A.D.) summer 850-hPa winds (unit: m/s) derived from NCEP/NCAR reanalysis (Kalnay et al., 1996); (b) Taylor diagram displaying the pattern statistics of climatological summer 850-hPa meridional winds over East Asia (20°–45°N, 105°–135°E) between PMIP3 historical experiments and observations. The radial coordinate is the standard deviation normalized by observations, and the angular coordinate is the spatial correlation with observations. The normalized centered root-mean-square error between a model and observation (marked as REF) is the distance between them. Note that the CSIRO-Mk3L-1-2 simulation has a negative spatial correlation with the observation and hence is not shown in (b).**





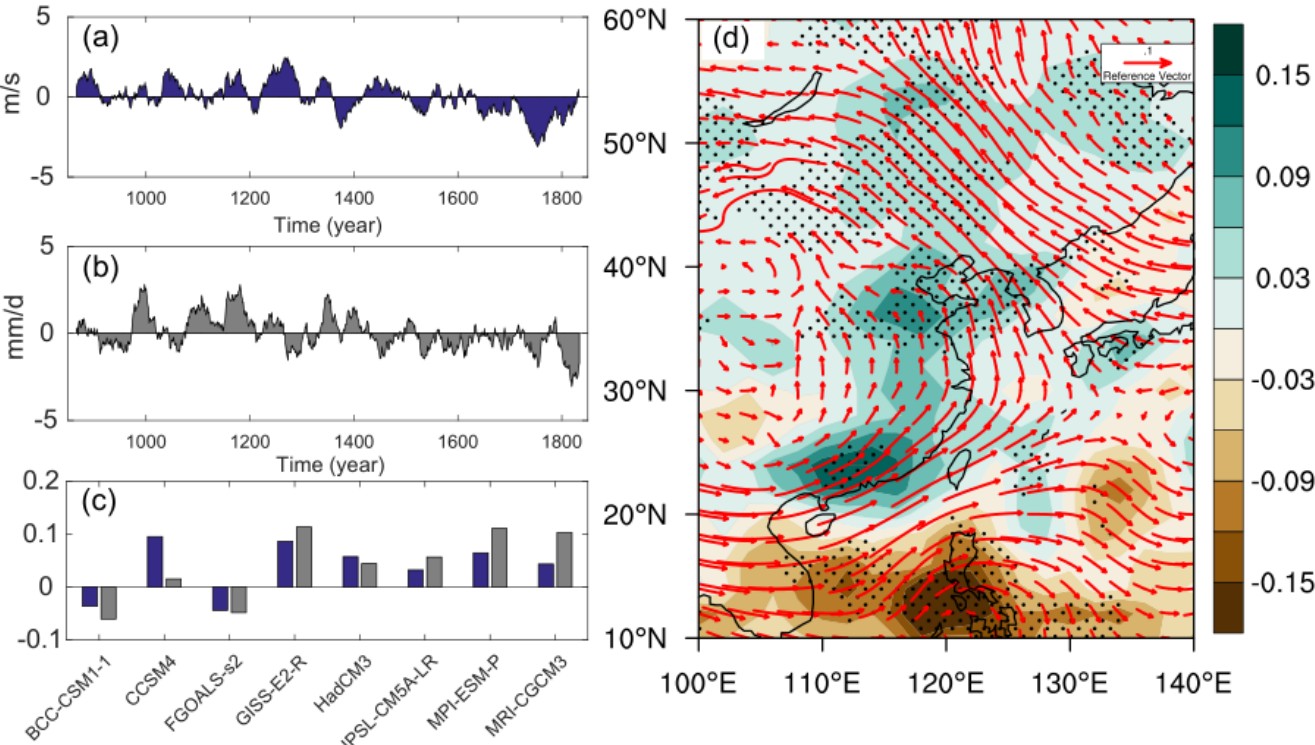

**Figure 2. (a) MEM of EASM strength anomalies relative to the long-term average (850–1850 A.D.); (b) same as (a) but for regionally averaged summer precipitation anomalies over eastern China (20°–45°N, 105°–120°E); (c) difference in the EASM strength (blue) and regionally averaged summer precipitation over eastern China (gray) between the MCA and LIA; (d) MEM of spatial changes in winds (vectors, in m·s⁻¹) and summer precipitation (shading, in mm·d⁻¹) between MCA and LIA. Dotted areas show that at least six of eight models have the same sign as for MEM.**





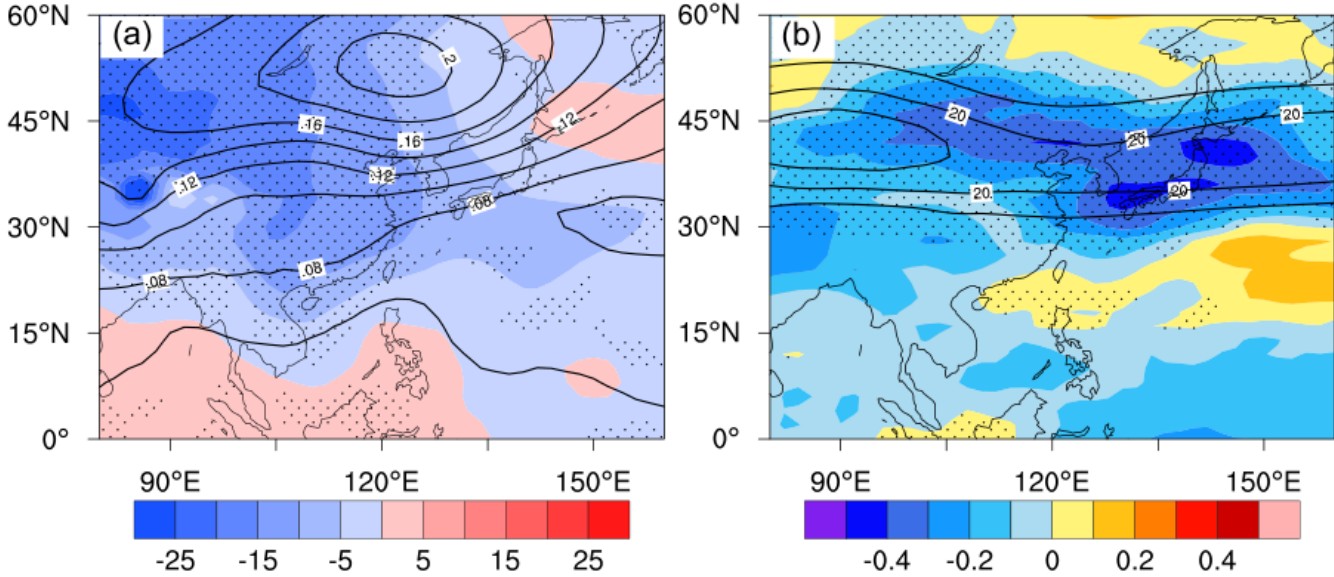

**Figure 3. (a) MME of differences in the summer surface air temperature (contours, in °C) and sea level pressure (shading, in Pa) between the MCA and LIA; (b) MME of differences in the summer 200-hPa zonal winds (shading, m·s⁻¹) between the MCA and LIA. Red contours are the climatological (850–1850 A.D.) 200-hPa zonal winds (≥ 15 m·s⁻¹, contour interval is 5 m·s⁻¹). Areas passing the 95% significance test are dotted.**





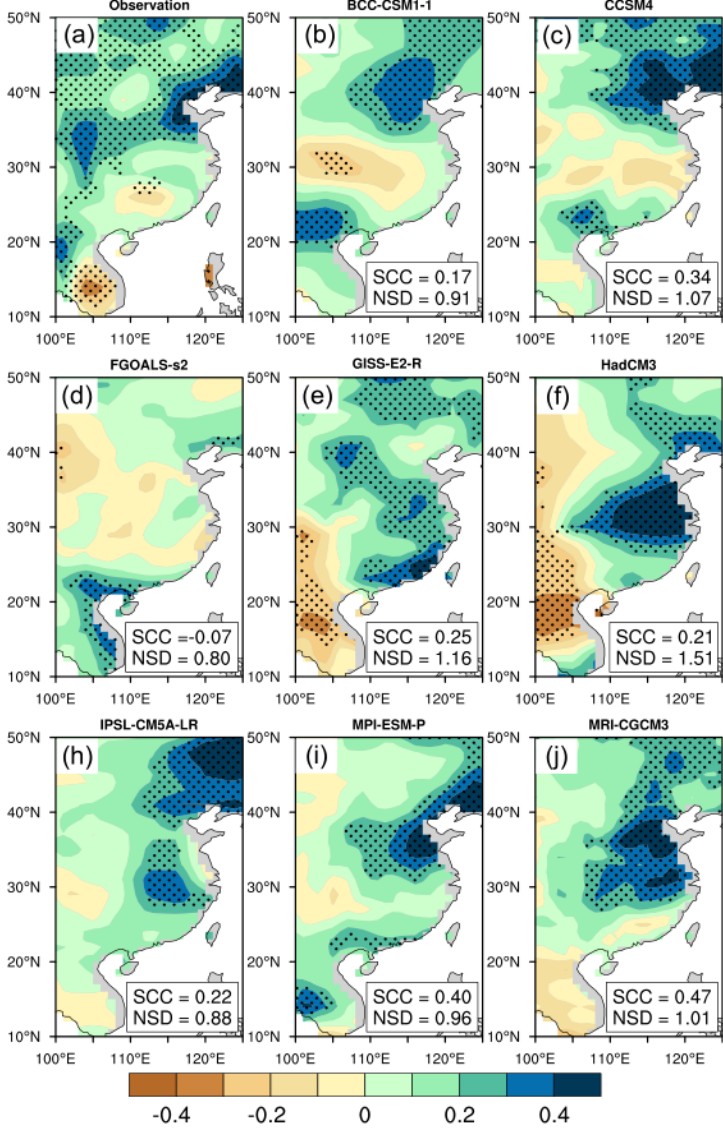

**Figure 4. Distribution of the correlation coefficients between the EASM strength and summer precipitation over East Asia in (a) observations (1979–2000 A.D.) and (b-j) PMIP3 historical experiments, with spatial correlation coefficients (SCCs) and normalized standard deviations (NSDs) between models and observations shown in bottom right corner. The observed EASM-precipitation relationship is calculated with NCEP/NCAR reanalysis (Kalnay et al., 1996) and Global Precipitation Climatology Project (GPCP) V2.3 dataset (Adler et al., 2003). Areas passing the 95% significance test are dotted.**




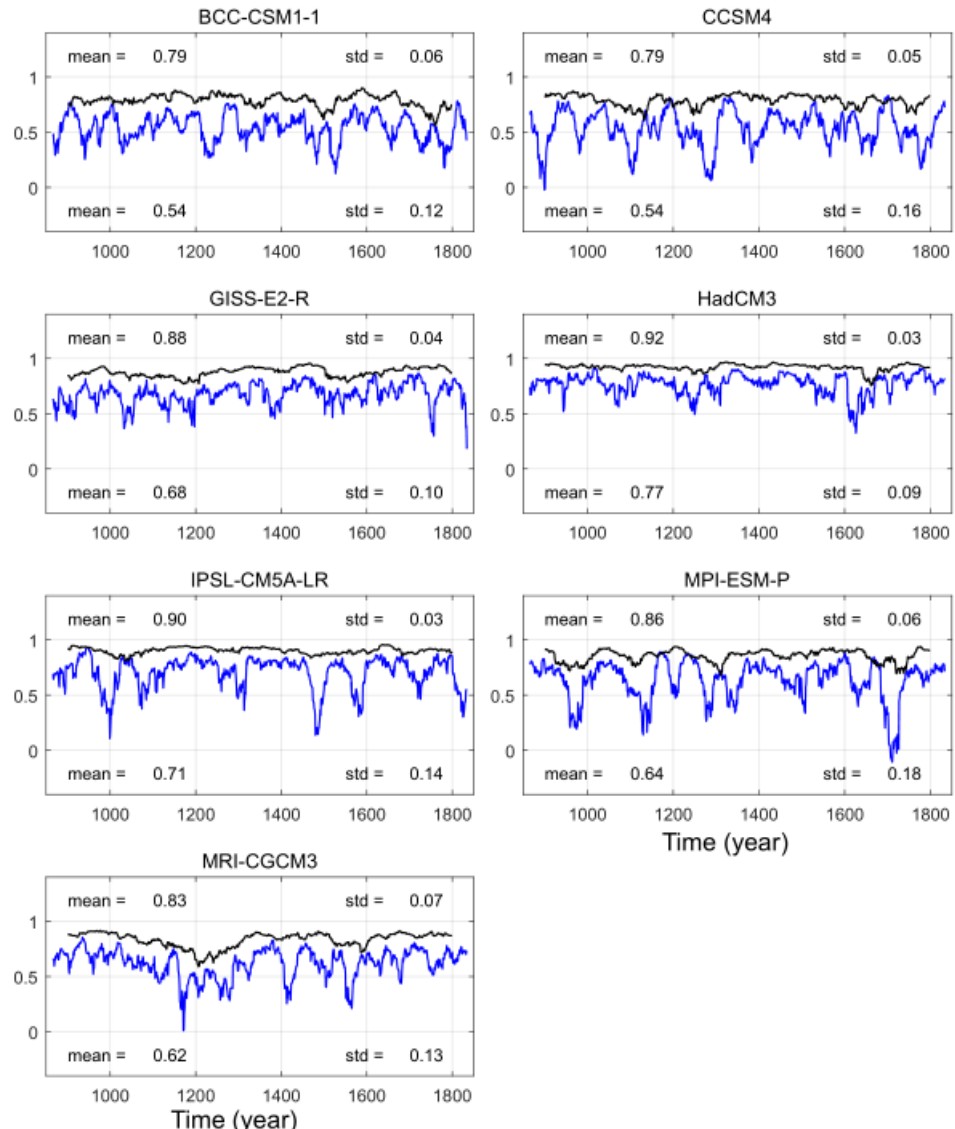

**Figure 5. Running pattern correlations (RPCs) of the EASM strength and summer precipitation over East Asia (10°–50°N, 100°–130°E) in the PMIP3 models. Blue (Black) lines represent 31-year (101-year) RPCs, with averages and standard deviations shown below (above).**





**Figure 6. Running correlations (RCs) between the EASM strength and regionally averaged summer precipitation over eastern China (25°–45°N, 105°–120°E). Blue (Black) solid lines represent the 31-year (101-year) RCs, with upper blue (lower black) dashed lines shown as 95% significance test values.**




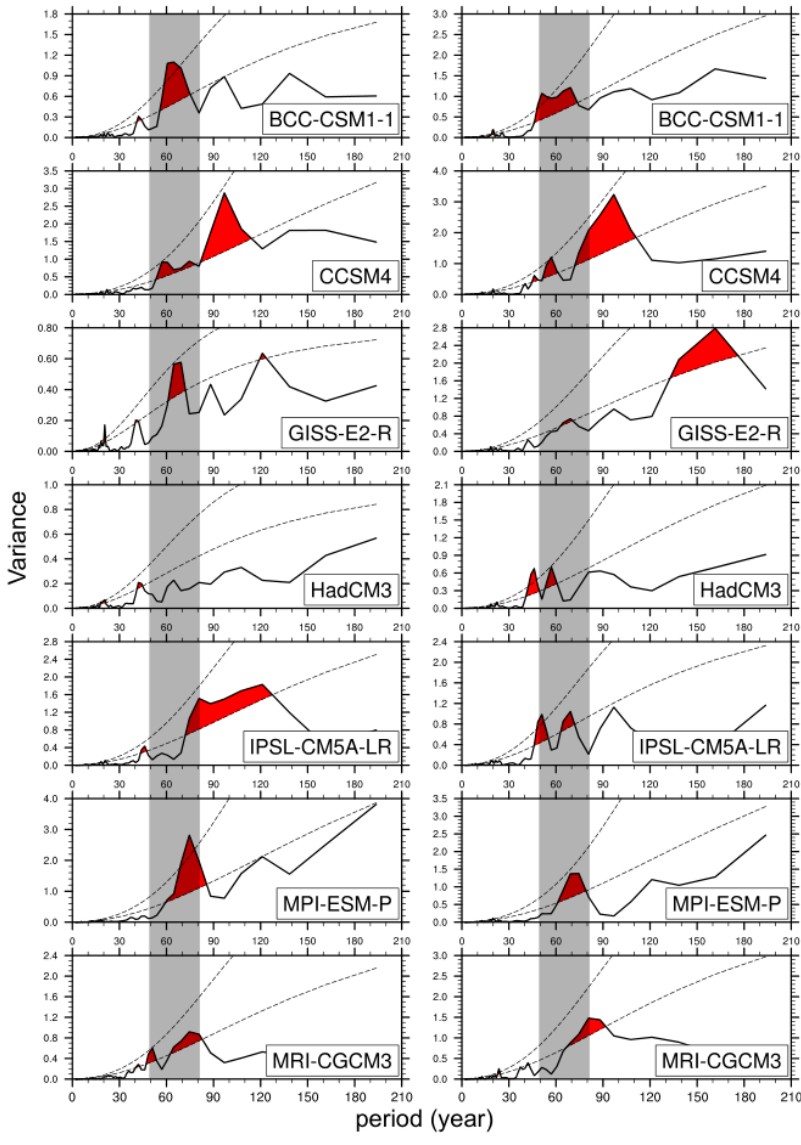

**Figure 7. Power spectrum of 31-year RPCs (left) and RCs (right) in the PMIP3 models. Lower (upper) dashed lines represent 90% (95%) significance test values, and power spectra passing 90% significance test are in red.**





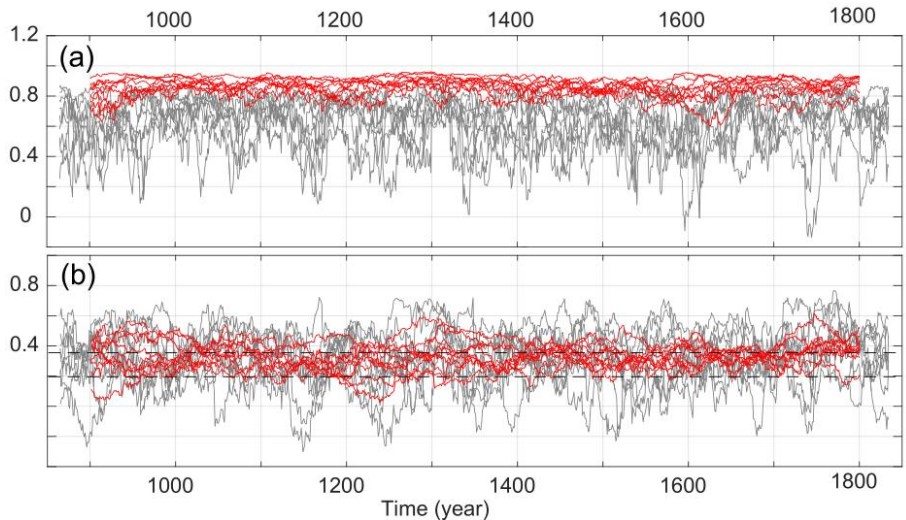

**Figure 8. (a) RPCs and (b) RCs in nine CESM-LME full-forcing experiments. Gray (Red) lines represent 31-year (101-year) RPCs/RCs. Dashed lines in (b) represent the 95% significance test for 31-year (upper) and 101-year (lower) RCs. Note that the RCs in (b) are calculated between the EASM strength and regionally averaged (35°–45°N, 105°–120°E) summer precipitation.**



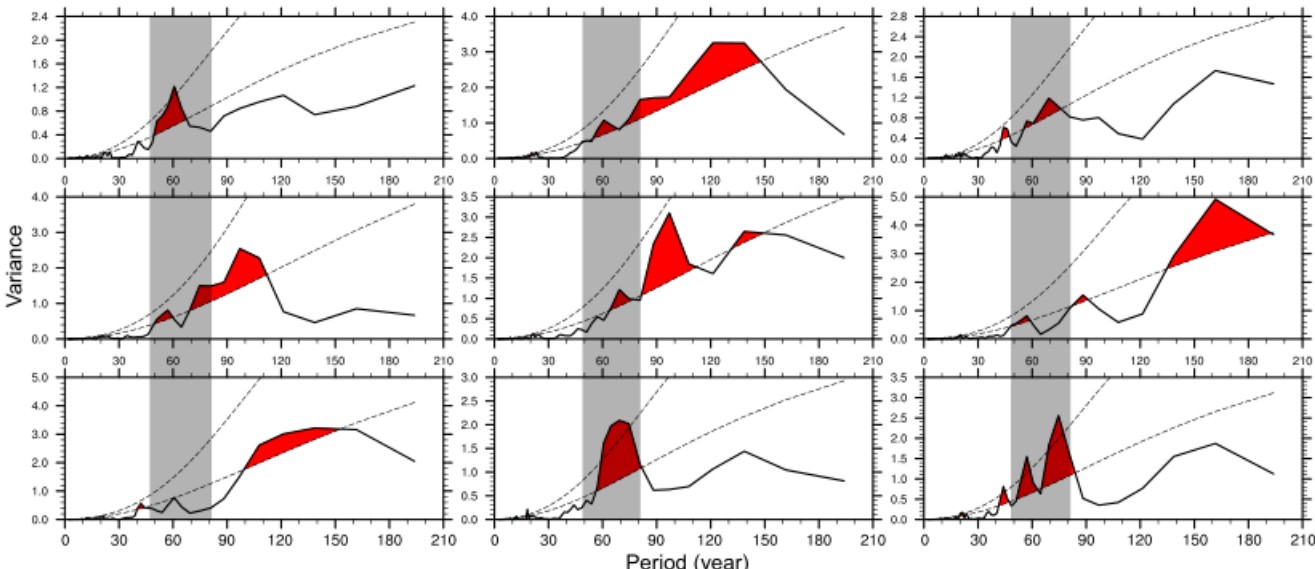

**Figure 9. Same as Fig. 7 but for 31-year RCs in nine CESM-LME full-forcing experiments.**





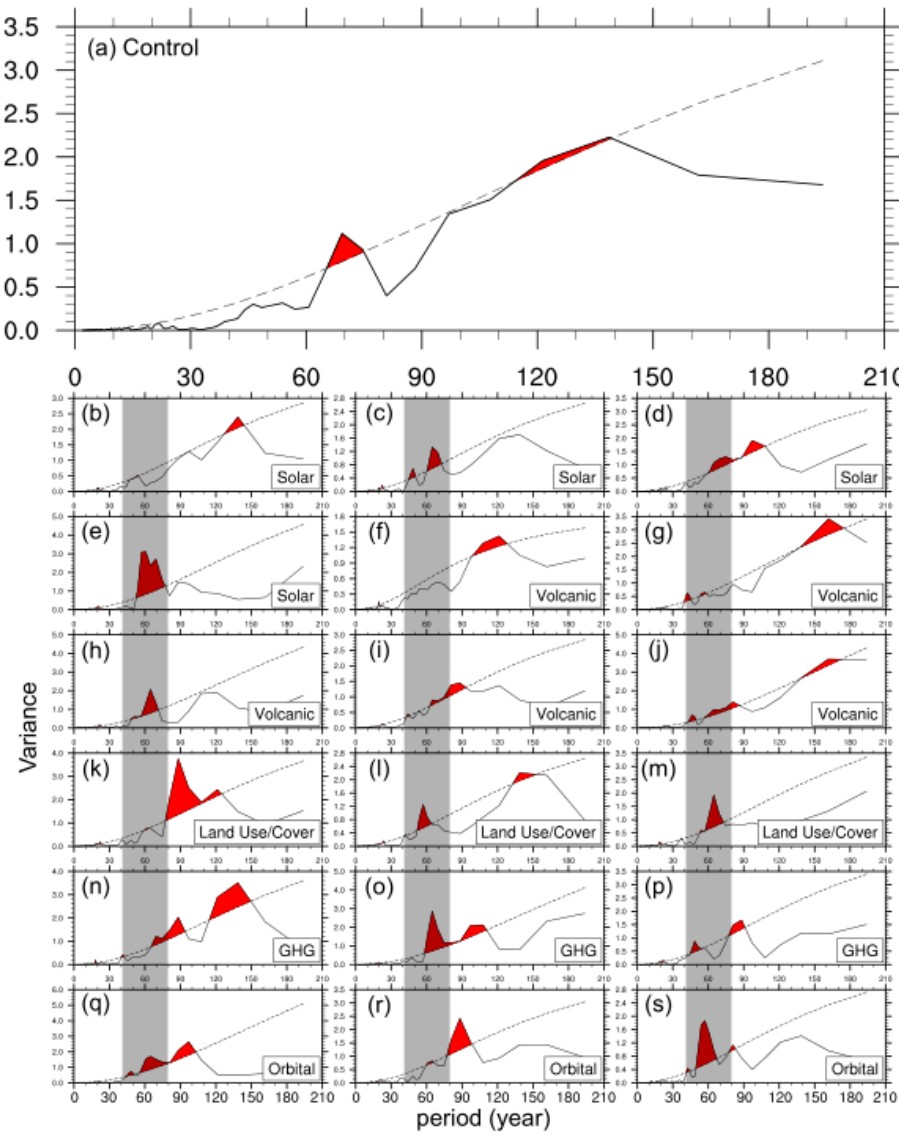

**Figure 10. Same as Fig. 9 but for 31-year RCs in CESM-LME (a) control experiment and (b–s) single-forcing experiments. The bottom right text represents the forcing applied in each single-forcing experiment. Dashed line represents 90% significance test values.**



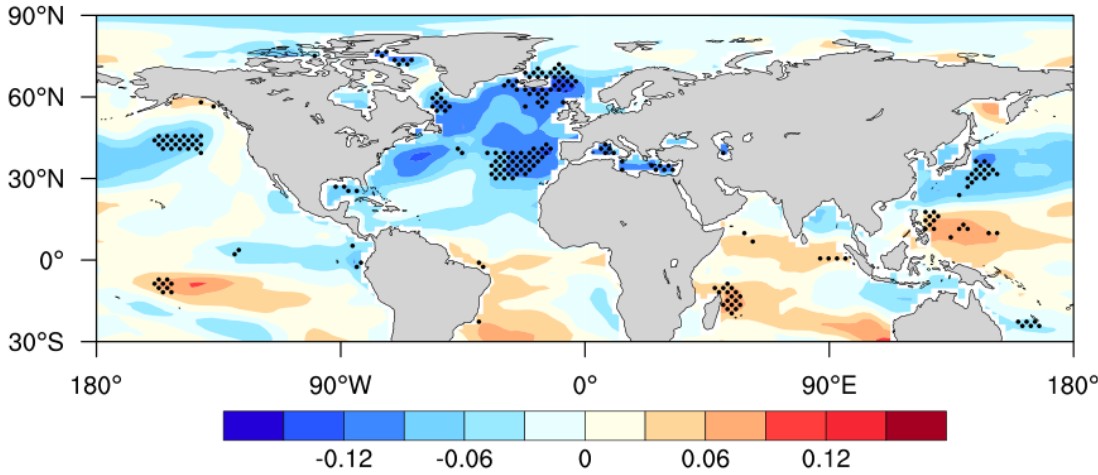

**Figure 11. MEM of correlation coefficients between 31-year RCs and 31-year running averaged summer SST anomalies among CESM-LME full-forcing experiments. Dotted areas show that at least seven of nine ensemble members have the same sign with MEM result.**