# Peer review of "Timescale-dependence of the relationship between the East Asian summer monsoon strength and precipitation over eastern China in the last millennium"

_Climate of the Past, 2017_

## Referee Comment (RC1) · Anonymous Referee #1 · 5 Nov 2017

The manuscript observed that proxy records representing seasonal precipitation over Eastern China dn and East Asia, believed to be driven by the intensity of the East Asian Summer Monsoon over the past millennium, tend to disagree at decadal and multi-decadal timescales, while agreeing at multi-centennial time scales . The authors attempt to explain this apparent inconsistency by analyzing the past millennium simulations of the PMIP3 model suite and in the Last Millennium Ensemble conducted with the CESM Earth System Model. The main conclusion is that the climate models exhibit a similar behaviour as the proxy records, with robust relationship between the

simulated EASM and simulated East sian precipitation, while showing inconsistent or high variable links between the ESM and precipitation at multidecadal timescales. The authors also conclude that the variability between EASM and precipitation is caused by internal climate dynamics - not related to the external climate forcing. In this regard, the sea-surface temperatures in the North Atlantic seem to play a role in this variable link.

In general terms, I think the manuscript is well written and its objectives are appealing. I am less satisfied with the last part of the manuscript, which includes the spectral analysis of the links between EASM and precipitation and the possible role of the North Atlantic SSTs. I think that this part is very speculative in less well supported than the rest. In my view this part requires some revisions, both considering the methods applied and possibly also the conclusions.

The English usage - for me a non native English speaker - also requires some light polishing.

Main points.

-The manuscript reads well and is convincing until section 4.2. In this section looks into the spectral characteristics of the correlation between the EASM and precipitation in the model simulations. It claims that there exist a 60-year quasi periodicity i almost all PMIP3 simulations and the CESM ensemble. I have two main concerns. One is that the spectra of the running correlations in the CESM ensemble do not really look similar in the different ensemble members. This ensemble has been conducted with the same forcing and with the same model, so that the spectra - if they represent a real signal- should look, in my opinion , much more similar. For instance, the simulation in left column middle row shows a spectrum that is very different from the simulation in the right column middle row. This mens that either the statistical significance of the spectral peaks is not really well estimated: the peak at about 130 years that appears in this latter simulation as significant does not appear in any other ensemble member.

This may be due to the construction of the time series. The running correlation are calculated with a 31-year filtered applied to the EASM and precipitation time series. I suspect that this filtering may introduce spurious peaks in the spectrum, although it is difficult ascertain before hand. I suggest to calculate the spectrum of a time series resulting from calculating the running correlation of random time series and see in how many cases spurious spectral peaks arise . I found a bit suspicious that the spectral peaks that the authors claim are twice and four times the period of the running window width.

-The explanation of the involvement of the North Atlantic SSTs on the link between EASM and precipitation is actually very weak. It is based on a statistical result without any physical explanation. This s a reflection of another weakness in the study. The authors clearly show that there are multi-decadal periods where the link EASM-preciptation breaks down. This must have a local and immediate reason, for instance that in those periods other local patterns of variability vary more strongly, or that the souces of moisture in the Western Pacific become colder or other similar reason. But if there is a long-distance effect of the North Atlantic SSTs, this has to be mediated by a regional mechanism, add this is not explored at all in the study. In addition, the correlations displayed in Figure 11 are really low. Tis figure also shows the area where at least 7 of the nine ensemble members show the same sign of the correlation. However, this result may not be that significant as it seems at first sight. On average 4 or 5 simulations will show the same sign, so that 7 can be not that unusual when considering that this test is applied to all grid cells of the simulation at the same time (this is the simultaneous multiple test problem or field significance). In other words, the chances that one single region in the world passes the 7-over-nine-same-sign test are probably not that low.

Minor points

3. Page 2, line 5: 'aforementioned EASM-precipitation relationship is possibly changeable over recent decades (e.g., Shi and Zhu, 1998; Li et'

Perhaps, changeable -> not stable

4. s. Peng et al. (2014) also implied that several severe droughts that occurred over eastern China were

Peng et al is not in the reference list

5. page 3, line 21 :'ESM because of its climate drift in long-term simulations (Gupta et al., 2013). These simulations have a rough time span '

rough time scape -> approximately cover a span

6. Page4, line2 :Specifically, we calculate the geological distributions of the correlation between the EASM strength and summer precipitation

geological -> spatial

7. Page 4, line 11:' CGCM3) to 0.79 (GISS-E2-R), all passing the 95% significance test. The centered root-mean-square errors range from 0.99 (MRI-CGCM3) to 1.55 (HadCM3),

are over the 95% significance level.

units for rmse are missing - I guess they are m/sec

---

## Referee Comment (RC2) · Anonymous Referee #2 · 17 Nov 2017

Overall rating: Although the paper addresses a relevant topic and uses adequate tools major revisions are necessary to provide a substantial contribution to Climate of the Past.

The manuscript investigates the timescale dependence of the relationship between the East Asian Summer Monsoon (EASM) and precipitation over eastern China using the results of the multi-model ensemble provided by PMIP3 and the single-model ensemble of last –millennium simulations from NCAR. The paper addresses a relevant question for paleoclimate research. The often assumed connection between the EASM strength

and Chinese precipitation variations is the basis for several monsoon reconstructions, e.g. from speleothems. Discrepancies between models and reconstructions and between different reconstructions may be rooted in the fact that the precipitation patterns over eastern China are influenced by other circulation features than the EASM. The authors look at the dependency on time-scales and find that the connection between EASM and a north-south precipitation pattern over eastern China is relatively stable over longer periods, such as the Medieval Climate Anomaly and the Little Ice Age. On shorter time-scales the relation breaks down and other mechanisms or remote drivers become likely more important for the regional precipitation distribution. Another important and robust conclusion of the analyses is that external forcing does not influence the shorter-timescale variations in the relationship between EASM and rainfall. These conclusions are based on a thorough analysis of the multi-model PMIP3 past1000 ensemble. The authors first establish the performance of individual models in representing present-day climate variations and base the selection of models on this evaluation. The results could be more robust if the athors included also the CESM single-model ensemble in the analyses of section 4.1 (see below). I am much less convinced about section 4.2, where the authors claim that a roughly 60yr oscillation in some teleconnections to the North Atlantic causes the variations in the EASM-precip relation. I don't find the collection of spectra very convincing and strongly recommend not to derive spectra from heavily-smoothed time series. The connection with the Atlantic Multidecadal Oscillation is also not well established. The correlations shown in Fig.11 show extremely low explained variance, even though they may pass a statistical significance test. In an earlier paper (Shi et al., Clim. Dyn., 2016) the principle author did a much better job in identifying teleconnections influencing precipitation patterns in a particular model. If the AMO-China precip connection is as robust as the authors claim, the multitude of realisations frim the CESM ensemble should make it possible to nail down the pathway how and to which amount the AMO influences the eastern China rainfall in comparison to the EASM.

Minor issues:

Page1, line 23 have == has Page 2, line18: do you mean "combing" or "combining"? Page 2, lines32ff and general: I recommend to reduce (increase) the wording in parenthesis in order to improve (deteriorate) the clarity of the argument. Page 3, line 5: Peng et al., 2014: for which time period? Line 6: the previous work by Shi et al is important and you should give a brief summary of their findings. Also for the later part: In Shi et al. 2016 it is concluded that only one model is able to adaequately reproduce the precipitation patterns over China in the last millennium context. Why is that not so important for the present study?

Line 21: the PMIP3 definition is exactly 850 to 1850 A.D.

Page 4, line 2, page 5, line 11: why geological? Geographical, spatial?

Page 5, line 11, figures 1,4: the "observations" are from a relatively short period (1979-2000). In the light of the later results on the non-stationarity: How does one know that this period is representative for the 20th century or longer?

Page 6, lines 6ff: The results could be made more robust if the CESM LM Ensemble simulations would be included. For example, in figure 2c one could have another entry for CESM LME including an estimate of the ensemble spread. So you would provide both a multi-model ensemble and a single-model ensemble.

Page 6, lines 14ff: I don't find the periodicity so obvious. If one requires 95% significance, only 5 out of 14 PMIP models and 3 out-of 9 LME simulations meet the criterion, hardly a very robust feature. Again, the spectra should not be calculated from smoothed data. Line 15, and page 7, line 13: There is only one Shi et al., 2016 in the reference list.

Line 29: I would say there are as significant peaks between 120 and 150 years in several of the individual forcing runs (e.g., 10 b, f,I,n)

Page 7, line 29: "geological evidence" better: from proxy data

---

## Author Comment (AC1) · 5 Jan 2018

**Responses to Reviewer#1**

1. The manuscript reads well and is convincing until section 4.2. In this section looks into the spectral characteristics of the correlation between the EASM and precipitation in the model simulations. It claims that there exist a 60-year quasi periodicity is almost all PMIP3 simulations and the CESM ensemble. I have two main concerns. One is that the spectra of the running correlations in the CESM ensemble do not really look similar in the different ensemble members. This ensemble has been conducted with the same forcing and with the same model, so that the spectra - if they represent a real signal- should look, in my opinion, much more similar. For instance, the simulation in left column middle row shows a spectrum that is very different from the simulation in the right column middle row. This means that either the statistical significance of the spectral peaks is not really well estimated: the peak at about 130 years that appears in this latter simulation as significant does not appear in any other ensemble member. This may be due to the construction of the time series. The running correlation are calculated with a 31-year filtered applied to the EASM and precipitation time series. I suspect that this filtering may introduce spurious peaks in the spectrum, although it is difficult ascertain before hand. I suggest to calculate the spectrum of a time series resulting from calculating the running correlation of random time series and see in how many cases spurious spectral peaks arise. I found a bit suspicious that the spectral peaks that the authors claim are twice and four times the period of the running window width.

Thank you very much for the insightful comments.
1. We acknowledge that there are some differences among the spectra derived from CESM ensemble members. On one hand, we focus on the spectra of RPC/PC, which depends not only on variations of the winds and precipitation but also on their relationship. This makes the spectra more complicated than a simple index, such as a precipitation index. On the other hand, the key factor affecting the fluctuation of the PRC/RC may be different among the CESM simulations. As you mentioned, the correlation shown in Fig. 11 is relatively low, though it is statistically significant. This result suggests that other factors except for the AMO may also contribute to the RC fluctuation, but they are not robust signs among the ensemble members. In other words, although the model employs the same external forcings, the temporal evolution of internal variability of climate system (e.g., PDO) is not the same due to different initial conditions. As listed in Table S1 and S2, the SST over North Pacific is more sensitive to the initial conditions than that over the North Atlantic in the CESM-LME simulations, resulting in the complex phase combinations of the AMO and PDO (Fig. S7). Therefore, the combination of different phase of internal variabilities may enhance or damp the relationship between the winds and precipitation, hence leading to the complexity of the spectra. In the revised manuscript, we add some discussions on this point (Page 8, Line 1-8).
2. Following your suggestions, we verified the significance of the spectrum with a Monte Carlo simulation (Page 4, Line 9-14). Specifically, we generated two random arrays with the same length of the original data, and then calculate the spectrum of their 31-year running correlation. We repeat previous steps 10,000 times and get 10,000 spectra, the fifth (tenth) percentile at each timescale is set as threshold for $\alpha = 0.05$ ($\alpha = 0.1$). As shown in Fig. 10, the running correlation between the original data could induce some spurious spectral peaks on short timescales (i.e., 22-year) but not on a longer timescale. Thus, in some ensemble members, the spectral that peaks around twice and four times of the running window width is unrelated to the application of running correlation.

2. The explanation of the involvement of the North Atlantic SSTs on the link between EASM and precipitation is actually very weak. It is based on a statistical result without any physical explanation. This s a reflection of another weakness in the study. The authors clearly show that there are multi-decadal periods where the link EASM-precipitation breaks down. This must have a local and immediate reason, for instance that in those periods other local patterns of variability vary more strongly, or that the sources of moisture in the Western Pacific become colder or other similar reason. But if there is a long-distance effect of the North Atlantic SSTs, this has to be mediated by a regional mechanism, add this is not explored at all in the study. In addition, the correlations displayed in Figure 11 are really low. This figure also shows the area where at least 7 of the nine ensemble members show the same sign of the correlation. However, this result may not be that significant as it seems at first sight. On average 4 or 5 simulations will show the same sign, so that 7 can be not that unusual when considering that this test is applied to all grid cells of the simulation at the same time (this is the simultaneous multiple test problem or field significance). In other words, the chances that one single region in the world passes the 7-over-nine-same-sign test are probably not that low.

1. We add some discussions on the possible mechanisms about how the AMO affected the EASM-precipitation relationship (Page 7, Line 23-31).

The formation of precipitation is not only affected by the moisture but also related to the local thermal condition. When the temperature gets lower, the moist air gets easier to saturate if the moisture is constant. Previous studies have shown that the AMO could influence the temperature over East Asia positively (Lu et al., 2006; Wang et al., 2013). During the cold (warm) phase of the SST anomalies over North Atlantic, the temperature over East Asia tends to be colder (warmer) (Fig. S6). As the EASM strengthens, the moisture transported to monsoon region increases, which is propitious to improve precipitation. Meanwhile, the lower (higher)-than-normal temperature condition over East Asia is helpful (unhelpful) to the saturation of the air and thus promote (hamper) the formation of precipitation, which results in a more (less) robust positive EASM-precipitation relationship.

[Figure]

Figure S6. The summer surface temperature anomalies during the (a) negative phase (NA-) and (b) positive phase (NA+) of SSTA over the North Atlantic (30°−70°N, 80°W−0°). (c) The difference in summer surface temperature between the NA- and NA+. The NA+ (NA-) is selected for the time periods that the summer SSTA over North Atlantic exceed its 1.2 (-1.2) standard deviation. Units: °C.

2. We applied the Monte Carlo simulations to demonstrate that the SST variation over the North Atlantic is a significant factor connecting with the EASM-precipitation relationship (Fig. xx?). Although PDO is another potential factor regulating the EASM-precipitation relationship, its temporal evolution is very sensitive to the initial conditions among the CESM-LME (Table S2), that is, the phase of PDO varies with individual ensembles in the same time interval. This may be responsible for the low variance explained by the AMO shown in Fig. 11.

3. Figure R1 shows that most anomalous SSTs around the world only passed the 4- or 5-over-9-same-sign-test, except over the North Atlantic region, where can passed the 7-over-9-same-sign-test. The result of 7-over-9-same-sign-test is similar to that of the Monte Carlo simulation test (Fig. 11), proving the rationality of the 7-over-9-same-sign-test.

[Figure]

Figure R1. Numbers of ensemble members in CESM-LME full-forcing experiments that agree with the MEM result in the connection between the AMO and EASM-precipitation relationship (shown in Fig. 11).

References

Wang, J., Yang, B., Ljungqvist, F. C., Zhao, Y.: The relationship between the Atlantic Multidecadal Oscillation and temperature variability in China during the last millennium, J. Quaternary Sci., 28, 653-658, 2013.

Lu, R., Dong, B., and Ding, H.: Impact of the Atlantic Multidecadal Oscillation on the Asian summer monsoon, Geophys. Res. Lett., 33, 2006.

3. Page 2, line 5: 'aforementioned EASM-precipitation relationship is possibly changeable over recent decades (e.g., Shi and Zhu, 1998; Li et'
Perhaps, changeable -> not stable
Corrected.

4. Peng et al. (2014) also implied that several severe droughts that occurred over eastern China were.
Peng et al is not in the reference list
We add the reference of Peng et al. (2014). (Page 11, Line 23-24)

5. page 3, line 21:'ESM because of its climate drift in long-term simulations (Gupta et al., 2013).

These simulations have a rough time span'
rough time scape -> approximately cover a span
Corrected.

6. Page4, line2: Specifically, we calculate the geological distributions of the correlation between the EASM strength and summer precipitation
geological -> spatial
Corrected.

7. Page 4, line 11:' CGCM3) to 0.79 (GISS-E2-R), all passing the 95% significance test. The centered root-mean-square errors range from 0.99 (MRI-CGCM3) to 1.55 (HadCM3),
are over the 95% significance level.
units for RMSE are missing - I guess they are m/sec
The RMSE has been normalized by the standard deviation of the observation, thus it is a dimensionless quantity.

---

## Author Comment (AC2) · 5 Jan 2018

**Responses to Reviewer#2**

1. These conclusions are based on a thorough analysis of the multi-model PMIP3 past1000 ensemble. The authors first establish the performance of individual models in representing present-day climate variations and base the selection of models on this evaluation. The results could be more robust if the authors included also the CESM single-model ensemble in the analyses of section 4.1 (see below). I am much less convinced about section 4.2, where the authors claim that a roughly 60yr oscillation in some teleconnections to the North Atlantic causes the variations in the EASM-precip relation. I don't find the collection of spectra very convincing and strongly recommend not to derive spectra from heavily-smoothed time series.

Thank you very much for the constructive comments.

1. We include the CESM-LME results in Page 6, Line 31-34, and attach the corresponding figure in the supplement (Fig. S3).

2. The multi-decadal periodicity of PRCs/PCs differs among different models. On one hand, the RPC/PC depends not only on variations of the winds and precipitation but also on their relationship. This makes their spectra more complicated than a simple index, such as a precipitation index. Thus, the different parameterizations relevant to precipitation and winds among PMIP3 models may lead to the diversity of their spectra. On the other hand, the key factor affecting the fluctuation of the PRC/RC may vary even among the same model simulations (i.e., CESM-LME). The correlation between the AMO and EASM-precipitation relationship is relatively low, though it is statistically significant (Fig. 11). This result suggests that other factors except for the AMO may also contribute to the RC fluctuation, while not robust among the ensemble members. Take the PDO for example, we show that the SST variation over the North Pacific is more sensitivity to the initial conditions than that over the North Atlantic (Table S1 and S2), making their phase combinations differ among CESM-LME (Fig. S7). In other words, the connection between the AMO and EASM-precipitation relationship is affected by the PDO in different ways among the CESM-LME, making the spectra of RC/PRCs more complex. We add some discussions on this point (Page 8, Line 1-8) in the revised manuscript. Considering these reasons, we modified the "roughly 60-year periodicity" into the "multi-decadal periodicity" throughout the manuscript.

3. It is possible that heavily-smoothed time series could induce spurious peaks in the spectrum. To avoid this problem, we applied a Monte Carlo simulation to verify the significance of the spectral. Specifically, we generated two random arrays with the same length of the original data, and then calculate the spectrum of their 31-year running correlation. We repeat previous steps 10,000 times and get 10,000 spectra, the fifth (tenth) percentile at each timescale is set as threshold for $\alpha = 0.05$ ($\alpha = 0.1$). As shown in Fig. 10, the running correlation will induce some spurious spectral peaks on the short timescales (i.e., 22-year) but not on the timescale we concerned (i.e., around 60-year). We introduce the Monte Carlo simulation in Page 4, Line 9-14.

2. The connection with the Atlantic Multidecadal Oscillation is also not well established. The correlations shown in Fig.11 show extremely low explained variance, even though they may pass a statistical significance test. In an earlier paper (Shi et al., Clim. Dyn., 2016) the principle author did a much better job in identifying teleconnections influencing precipitation patterns in a particular model. If the AMO-China precip connection is as robust as the authors claim, the multitude of realisations from the CESM ensemble should make it possible to nail down the pathway how and

to which amount the AMO influences the eastern China rainfall in comparison to the EASM.

We add some discussion on the possible mechanisms about the possible mechanisms that AMO affected the EASM-precipitation relationship (Page 7, Line 23-31). However, it is difficult to distinguish the roles of AMO and EASM in precipitation over eastern China, as AMO also affects the EASM via modulating large-scale circulations (Lu et al., 2006; Yang et al., 2017).

The formation of precipitation is not only affected by the moisture but also related to the local thermal condition. When the temperature gets lower, the moist air gets easier to saturate if the moisture is constant. Previous studies have shown that the AMO could influence the temperature over East Asia positively (Lu et al., 2006; Wang et al., 2013). During the cold (warm) phase of the SST anomalies over North Atlantic, the temperature over East Asia tends to be colder (warmer) (Fig. S6). As the EASM strengthens, the moisture transported to monsoon region increases, which is propitious to improve precipitation. Meanwhile, the lower (higher)-than-normal temperature condition over East Asia is helpful (unhelpful) to the saturation of the air and thus promote (hamper) the formation of precipitation, which results in a more (less) robust positive EASM-precipitation relationship.

[Figure]

Figure S6. The summer surface temperature anomalies during the (a) negative phase (NA-) and (b) positive phase (NA+) of SSTA over the North Atlantic (30°−70°N, 80°W−0°). (c) The difference in summer surface temperature between the NA- and NA+. The NA+ (NA-) is selected for the time periods that the summer SSTA over North Atlantic exceed its 1.2 (-1.2) standard deviation. Units: °C.

References:

Yang, Q., Ma, Z., Fan, X., Yang, Z. L., Xu, Z., and Wu, P.: Decadal modulation of precipitation patterns over eastern China by sea surface temperature anomalies, J. Climate, 30, 7017-7033, 2017.

Lu, R., Dong, B., and Ding, H.: Impact of the Atlantic Multidecadal Oscillation on the Asian summer monsoon, Geophys. Res. Lett., 33, 2006.

Minor issues:

1. Page1, line 23 have == has Page 2, line18: do you mean "combing" or "combining"?

Corrected. We mean "combining".

2. Page 2, lines32ff and general: I recommend to reduce (increase) the wording in parenthesis in

order to improve (deteriorate) the clarity of the argument.

We modified the manuscript as you suggested. (Page2, Line 19-21)

3. Page 3, line 5: Peng et al., 2014: for which time period?

During the last millennium (Page 3, Line 6).

4. Line 6: the previous work by Shi et al is important and you should give a brief summary of their findings.

We add a brief summary of Shi et al. (2016b). (Page 3, Line 6-10)

5. Also for the later part: In Shi et al. 2016 it is concluded that only one model is able to adequately reproduce the precipitation patterns over China in the last millennium context. Why is that not so important for the present study?

In Shi et al. (2016a), they mainly focused on the divergent response of annual precipitation/humidity condition between arid Asian inland and Eastern China (monsoonal region), and select the models based on the annual humidity difference between the MCA and LIA in comparison with multi-proxies. In this study, we concentrated on the relationship between summer precipitation and winds over China. We choose models based on their performance in simulating the EASM strength and EASM-precipitation relationship, which is difficult to compare with the proxies. Thus, in Shi et al. (2016a), their model selection was stricter than that in this study. In addition, their result further agreed with one of our main conclusions that the EASM-precipitation relationship is positive and stable on multi-centennial timescale.

6. Line 21: the PMIP3 definition is exactly 850 to 1850 A.D.

Corrected.

7. Page 4, line 2, page 5, line 11: why geological? Geographical, spatial?

Corrected to "spatial"

8. Page 5, line 11, figures 1,4: the "observations" are from a relatively short period (1979- 2000). In the light of the later results on the non-stationarity: How does one know that this period is representative for the 20th century or longer?

We modified the observational EASM and EASM-precipitation relationship calculated from 1948-2000, and the simulated EASM and EASM-precipitation relationship are derived from 1901-2000. (Fig. 1 and 4; Fig S2)

9. Page 6, lines 6ff: The results could be made more robust if the CESM LM Ensemble simulations would be included. For example, in figure 2c one could have another entry for CESM LME including an estimate of the ensemble spread. So you would provide both a multi-model ensemble and a single-model ensemble.

Thanks for your suggestion and we added CESM results in the revised manuscript (Page 6, Line 31-33) and corresponding figure in the supplement (Fig. S3).

10. Page 6, lines 14ff: I don't find the periodicity so obvious. If one requires 95% significance, only

5 out of 14 PMIP models and 3 out of 9 LME simulations meet the criterion, hardly a very robust feature. Again, the spectra should not be calculated from smoothed data.

First, as you suggested, we applied a Monte Carlo simulation to avoid the possible spurious spectral peaks caused by filtered time series. We acknowledge that in some of these simulations, the spectral peaks are significant at 90% significance level, not robust compared to the 95% significance level. Nevertheless, almost all multi-decadal spectral peaks among these simulations pass the 90% significance test, which possibly indicates that it is a common and robust sign among climate models, thus increasing its credibility to some extent.

11. Line 15, and page 7, line 13: There is only one Shi et al., 2016 in the reference list.
References Shi et al. (2016a) and Shi et al. (2016b) are added. (Page 11, Line 32- Page 12, Line 2)

12. Line 29: I would say there are as significant peaks between 120 and 150 years in several of the individual forcing runs (e.g., 10 b, f,I,n)
As shown in Fig. 10, the CESM control run has a significant 120~150-year periodicity besides the multi-decadal periodicity. It indicates that the similar 120~150-year occurring in some CESM single-forcing runs may also result from the internal variability of the climate system. The lack of the 120~150-year periodicity in the remaining single-forcing runs possibly implicates that this periodicity is sensitive to the initial condition. We add some discussions on this point in the revised manuscript (Page 7, Line 4-7 and Line 17-18).

13. Page 7, line 29: "geological evidence" better: from proxy data
Corrected.

---

## Referee Report (RR1)

Review Shi

Overall rating: The authors have responded to the reviewers' comments and suggestions in an adequate way and have improved substantially the manuscript. In particular, they have provided better information on the statistical methods and have tuned down a bit their conclusion on a connection between their correlation indices to the Atlantic Multidecadal Variability. Rather, they make a clear and well-funded statement that the variations in the EASM/precipitation is an expression of internal variability, not a result of external forcing. This is an important finding for the interpretation of proxy records of different kinds. I recommend that the paper should be accepted for publication in CP after some minor revisions.

The authors have included more information from the CESM Last Millennium Ensemble (LME). However, this could be extended a bit in discussing the uncertainty regarding the single model results. For example, Figure 2c should be discussed further in connection with Figure S3 (which shows the same thing as far as I understood). In the LME runs, (fig. S3) the individual members show dramatic differences (3-4 with positive sign, 4-5 with negative sign. If such a distribution was representative for ensembles from the other models, one would have to conclude that the findings from individual runs (Fig. 2c) could be just by chance. I would also suggest providing a Taylor diagram figure in the supplement (as Fig. 1b) for the CESM LME.

Minor issues:
Page 3, ln 31: better: nine CESM-LME full-forcing experiment, one control experiment, and several sensitivity experiments with individual forcing (…

Page 6, ln 24ff: the "obvious" depends a bit on the view of the reader. In many simulations the 100-200 year periods are much more prominent that the 40-60 years. The authors play the lower-frequency a bit down, put it is very prominent when looking just at the RPC time series from, e,g, MPI-ESM-P.
Page 7, ln 15ff. along the same line: the 120-150 year period is prominent both in the inforced control run and the MME. Is there any evidence from other analyses of the LME where this comes from?

Page 8, ln 1ff: The fact that AMO is more consistent among ensemble members implies that AMO is more directly nfluenced by external forcing. This issue is discussed in a manuscript on a recent reconstruction of the AMO (Wang et al Nature Geosciences, 2017, doi:10.1038/ngeo2962), which could be included here.

---

## Author Response (AR2)

**Responses to Reviewer#1**

I thank the authors for considering my comments and for revising the manuscript accordingly. However, I still have one concern regarding the connection between the North Atlantic and the variations in the link between Monsoon and East Asia precipitation, as I explain below.

The authors do include a physical explanation to explain this link, via the modulation of temperature in East Asia by the North Atlantic SSTs. In their Fig S3 they show the composite fields of Asian temperature stratified by phases of the North Atlantic SST. Colder periods in East Asia should cause stronger precipitation-Monsoon link due to more effective condensation. In this chain of reasoning, I think there is a missing link, though. The authors should show in the first place that the air temperature in East Asia temperature is indeed the driving factor. This could be shown by calculating similar composite patterns as in Fig S3 but stratified by the precipitation-Monsoon links, i.e. constructing means of Asian temperature i periods where the precipitation-Monsoon link are weaker or stronger. Alternatively, they could show a figure similar to Fig11 but also showing the temperature over land in East Asia or over whole Asia for that matter. If the authors' hypothesis is true, we should see a clear correlation to air temperature.

[Figure]

**Figure 12. The summer surface temperature anomalies during the (a) high (RC+) and (b) low (RC-) EASM-precipitation relationship intervals in the CESM-LME full-forcing experiments; (c) and (d) are the same as (a) and (b), but for the clod (NA-) and warm (NA+) phases of summer SSTs over North Atlantic, respectively. The RC+ (NA+) and RC- (NA-) are selected for the periods that the EASM-precipitation relationship (summer SSTs over North Atlantic) exceed its 1.2 and -1.2 standard deviation, respectively. Units: °C**

As you suggested, we add Fig. 12 to show the connection between the EASM-precipitation relationship (RC+) and East Asian temperature (EAT). When the summer EAT is lower than normal, the RC tends to be closer (Fig. 12a), which could be explained by the temperature-condensation mechanisms proposed in Page 7, Lines 29-31. However, the temperature anomalies during periods of RC- (Fig. 12b) is not exactly opposite to that during the RC+ periods. This result suggests that the linkage between the EAT and RC is not simply linear, which further demonstrates the connection between the North Atlantic SSTs and EASM-precipitation relationship is very complicated. In the revised manuscript, we add some discussions on this point (Page 7, Line 30-34).

**Responses to Reviewer#2**

The authors have included more information from the CESM Last Millennium Ensemble (LME). However, this could be extended a bit in discussing the uncertainty regarding the single model results. For example, Figure 2c should be discussed further in connection with Figure S3 (which shows the same thing as far as I understood). In the LME runs, (fig. S3) the individual members show dramatic differences (3-4 with positive sign, 4-5 with negative sign. If such a distribution was representative for ensembles from the other models, one would have to conclude that the findings from individual runs (Fig. 2c) could be just by chance. I would also suggest providing a Taylor diagram figure in the supplement (as Fig. 1b) for the CESM LME.

We add some discussions on the uncertainties of the models results (Page 9, Line 6-11).

Although we use large amounts of climate simulations, uncertainties are inevitable. For example, PMIP3 simulations have robust signs in the climate anomalies between the MCA and LIA (Fig. 2), while the CESM-LME results vary largely among individual experiments (Fig. S3). The CESM simulations are driven by the same forcings and are only different in initial conditions. That is to say, the roles of external forcings are sensitive to initial conditions at least in the CESM-LME. It further implies that the conclusion based on the PMIP3 may just be a coincidence, while it is difficult to validate in the present study. Therefore, it is necessary to use more PMIP3 single model runs with different initial conditions to confirm the hypothesis.

In addition, we add a Taylor diagram figure (Fig. S3) to evaluate the performance of CESM-LME members to reproduce the modern EASM and EASM-precipitation relationship.

Minor issues:
Page 3, ln 31: better: nine CESM-LME full-forcing experiment, one control experiment, and several sensitivity experiments with individual forcing (…

Modified (Page 3, ln 31).

Page 6, ln 24ff: the "obvious" depends a bit on the view of the reader. In many simulations the 100-200 year periods are much more prominent that the 40-60 years. The authors play the lower-frequency a bit down, put it is very prominent when looking just at the RPC time series from, e,g, MPI-ESM-P.

We acknowledge that the centennial periodicities are also neglected in several simulations. First, the purpose of this study is to discuss the origins of mismatching among reconstructions on short-timescales, thus we pay more attention to their multi-decadal fluctuation. Second, the centennial periodicities are not as common as the multi-decadal periodicities among simulations, thus we thought it may not be a robust sign. Based on the above considerations, we mainly focus on the 40-60 years periodicities.

Page 7, ln 15ff. along the same line: the 120-150 year period is prominent both in the inforced control run and the MME. Is there any evidence from other analyses of the LME where this comes from?

Thank for pointing this issue out. We note that the 120-150 year period appears not only in the control run and but also in full-forcing runs ensemble, implying that this cycle is possibly induced by both internal and external forcings, and the role of external forcings may be more important.

According to Fig. S5, this cycle appears in almost all single forcing members expect for the orbital experiments. However, it is hard to say that these periods are caused by the corresponding forcings, because the internal variabilities of the climate system are not removed. Moreover, the 120-150 period is not always prominent in the experiments driven by a same forcing. One likely reason is that the external and internal forcings both influence the EASM-precipitation relationship, but their combinational effects vary largely among members. This issue increases the uncertainties of this study and is beyond the scope of present study. We will explore it in our future work.

Page 8, ln 1ff: The fact that AMO is more consistent among ensemble members implies that AMO is more directly nfluenced by external forcing. This issue is discussed in a manuscript on a recent reconstruction of the AMO (Wang et al Nature Geosciences, 2017, doi:10.1038/ngeo2962), which could be included here.

We add the reference of Wang et al. (2017). (Page 13, Line 15-16)